# Evaluating a co-designed care bundle to improve patient safety at discharge from adult and adolescent mental health services (SAFER-MH and SAFER-YMH): protocol for a non-randomised feasibility study

Natasha Tyler  ,[1,2] Ioannis Angelakis,[3] Richard Neil Keers,[1,4] Claire Planner,[1] Alexander Hodkinson,[5] Sally J Giles,[1] Andrew Grundy,[6] Navneet Kapur,[1,7] Chris Armitage,[1,8] Tom Blakeman,[1,2] Stephen M Campbell,[1] Catherine Robinson,[6] Jessica Leather,[1] Maria Panagioti[1,2]

For numbered affiliations see end of article.

**Correspondence to**
Dr Natasha Tyler;
natasha.tyler@manchester.ac.uk

## ABSTRACT

**Introduction** Patients being discharged from inpatient mental wards often describe safety risks in terms of inadequate information sharing and involvement in discharge decisions. Through stakeholder engagement, we co-designed, developed and adapted two versions of a care bundle intervention, the SAFER Mental Health care bundle for adult and youth inpatient mental health settings (SAFER-MH and SAFER-YMH, respectively), that look to address these concerns through the introduction of new or improved processes of care.

**Methods and analysis** Two uncontrolled before-and-after feasibility studies, where all participants will receive the intervention. We will examine the feasibility and acceptability of the SAFER-MH in inpatient mental health settings in patients aged 18 years or older who are being discharged and the feasibility and acceptability of the SAFER-YMH intervention in inpatient mental health settings in patients aged between 14 and 18 years who are being discharged. The baseline period and intervention periods are both 6 weeks. SAFER-MH will be implemented in three wards and SAFER-YMH in one or two wards, ideally across different trusts within England. We will use quantitative (eg, questionnaires, completion forms) and qualitative (eg, interviews, process evaluation) methods to assess the acceptability and feasibility of the two versions of the intervention. The findings will inform whether a main effectiveness trial is feasible and, if so, how it should be designed, and how many patients/wards should be included.

**Ethics and dissemination** Ethical approval was obtained from the National Health Service Cornwall and Plymouth Research Ethics Committee and Surrey Research Ethics Committee (reference: 22/SW/0096 and 22/LO/0404). Research findings will be disseminated with participating sites and shared in various ways to engage different audiences. We will present findings at international and national conferences, and publish in open-access, peer-reviewed journals.

## STRENGTHS AND LIMITATIONS OF THIS STUDY

⇒ This is a best practice intervention for making discharge from acute inpatient mental health settings to the community safer and more patient centred.
⇒ The study design, management and data analysis benefit from the input of experts by experience.
⇒ An in-depth mixed-methods approach (qualitative and quantitative) will be used to evaluate the study and inform the feasibility and key outcomes of the main trial.
⇒ A potential limitation of this study is the lack of a control condition, although an extensive baseline phase is applied for comparing data before and after introducing the intervention.

## INTRODUCTION

It is estimated that 1 in 4 adults and 1 in 10 children will be affected by mental or neurological disorders at some point in their lives.[1 2] Mental illness has become the single largest cause of disability in England, costing the National Health Service (NHS) over £13.3 billion in 2019/2020 alone and is rising among both children and adults.[3–5] Individuals who are admitted to inpatient mental health services are at higher risk of adverse outcomes such as shorter life expectancy; increased risk of homelessness; self-harm, violence and criminal activity; and increased likelihood of many other adverse psychosocial outcomes, including loneliness, isolation and stigma.[6–10]

Transitions into and out of acute psychiatric hospitals are risky stages in the healthcare pathway associated with patient safety

issues. The absence of continuity of care is an important risk factor for patient safety incidents during transitions of care.[8] Other patient safety risk factors may include: difficulties of medication management across the care system; delayed discharge often arising from insufficient housing, social or community care provision; and lack of information sharing between services (primarily between primary and secondary care).[6 10–14]

Multifaceted interventions have been increasingly used to improve patient safety and reduce readmissions in complex care transitions.[11 14–17] The SAFER patient flow bundle is an exemplar multifaceted intervention, developed by NHS England/Improvement to improve discharge from acute hospitals to the community.[18] The SAFER patient flow bundle consists of five components: (1) senior review (before midday); (2) expected discharge date and clinical criteria for discharge; (3) early assessments to improve patient flow; (4) early discharge (aiming to discharge patients before midday) and (5) a multidisciplinary review for patients with increased length of stay.[18] Many of the individual components of the SAFER bundle are used as standard practice or best practice guidelines in policy.[19] Preliminary evidence obtained by case studies across the country is promising as it shows that SAFER has resulted in reduced length of patient stay in different hospitals, diminished discharge delays with minimal complications and decreased readmissions or contact with primary care. Furthermore, patient and staff satisfaction have increased.[20]

Although protocols, webinars, and/or other tools have been developed to facilitate the implementation of SAFER,[20] there is currently lack of evidence examining its appropriateness within a mental health setting. To address this gap, we conducted a series of co-design empirical studies with multiple stakeholders including patients, carers, professionals and academics.[21 22] These led to the development of two adapted versions of the SAFER patient flow bundle tailored to mental health, one for an adult acute psychiatric hospital (SAFER-MH) and one for youth (child and adolescent) mental health services (SAFER-YMH). Now that the bundle has been developed, we aim to generate data regarding its feasibility and acceptability, which will enrich the evidence base about its future wider adoption in NHS.

The primary objectives of this research are therefore:
1. To examine whether the SAFER-MH and SAFER-YMH bundles are feasible and acceptable (based on standardised quantitative measures and qualitative data) to patients, carers and professionals.
2. To identify specific principle(s) of SAFER-MH and SAFER-YMH, which need to be adapted to make it more acceptable, and increase its prospect of long-term implementation.
3. To examine the adherence to the delivery of the intervention (measured using a purposefully designed self-report form and recruitment rates), completion rate and appropriateness of the outcome measures and recruitment strategies.

4. To explore whether healthcare professional and patient engagement with the SAFER-MH and SAFER-YMH interventions can be increased using evidence-based behavioural science techniques.

## METHODS AND ANALYSIS
### Study design
This study will include two concurrent distinct studies, receiving adapted versions of the same intervention:
► Phase 1: SAFER-Mental Health (MH) for adults.
► Phase 2: SAFER-Youth Mental Health (YMH) for children and adolescents, which includes an adaption of the original SAFER-MH for young people.

These will be uncontrolled before-and-after feasibility studies where all participants will receive the intervention. We will use quantitative (eg, questionnaires, completion forms) and qualitative (eg, interviews, process evaluation) methods to assess acceptability and feasibility.

### Intervention
The SAFER-MH is a co-designed adapted version of the SAFER patient flow bundle (NHS Improvement) intervention[18] and will be delivered to patients as part of their normal care pathway. The intervention was adapted based on 35 stakeholder interviews and RAND consensus methods with multidisciplinary experts and professionals.[21 22] SAFER-YMH involved a further level of co-design to ensure that the intervention is suitable for adolescent services. This involved presenting the adult version to professional and patient groups to discuss relevant changes to be made to make it more appropriate to adolescent services.

First, we aim to identify the elements of existing practice in the pilot sites before implementing the SAFER interventions, in both studies. The intervention focuses on promoting best practice guidance and has three key stages: admission, discharge and weekly tasks, and are discussed below:

### Admission
At admission, the intervention group will complete three key tasks: (1) setting criteria for discharge and an estimated discharge day, (2) identifying early social information that will help plan for discharge and (3) introducing the patient written discharge plan. This will be structured within two documents including the admission's social information capture document and the transition's checklist, which will be completed at multiple times in the patient journey.

### Weekly tasks
There are three weekly tasks that will form part of this assessment: (1) senior review of discharge readiness, (2) multidisciplinary discharge team meeting and (3) multiagency discharge team meeting.

## Discharge

At discharge, there will be two key tasks: (1) co-producing a high-quality patient written discharge plan and (2) ensuring the patient has their copy of the patient written discharge plan.

## Participants

All patients admitted to pilot wards during the study period (14 weeks from November 2022 onwards) will receive the intervention, as it is a best practice intervention delivered as a service improvement change in care. Each individual participant who is due to be discharged from the pilot ward will then be given the option to consent (parental/guardian consent for under 16 years) whether to be a participant in the research studies by participating in the interviews and by completing relevant questionnaires. Patient participants will be adults (phase 1) or young people (phase 2) with mental illness who were admitted to an inpatient ward. Healthcare professionals who will be delivering the intervention and carers/family members of the patients (who consent to be involved) will also be involved in the interview components of the study (patient and carer interviewees can decide whether to be interviewed alone or together).

We will collect data from all consenting patients who are discharged during the baseline and intervention stages, irrespective of whether their inpatient stay began before this period. In the case of an unplanned or early discharge, we will still collect data from consenting patients who have received the intervention; if a patient is not discharged from the pilot ward (due to transfer, etc), they will be ineligible to participate.

## Eligibility criteria
### Eligible sites

Eligible sites will be acute adult inpatient mental health wards. As this is a feasibility study, we will aim to test on a variety of wards (ie, female, male or mixed).

### Phase 1: SAFER-MH

The eligibility criteria for individual patient participants to take part in the SAFER-MH study reflect the eligibility criteria for services. We will include adult patients aged 18–65 years, informal carers for adult patients with mental health problems aged over 18 years, healthcare professionals who are involved in delivering the intervention and informal carers of patients (for process evaluation). All patients will receive the service-level intervention but will choose to participate in the study if they feel they have the capacity and consent.

### Phase 2: SAFER-YMH

The eligibility criteria to take part in the feasibility study reflect the eligibility criteria for services. Any young person accessing inpatient Child and Adolescent Mental Health Services (CAMHS) would be eligible to participate, as well as a parent/carer of a young person accessing inpatient CAMHS and healthcare professionals who are involved in delivering the intervention and parents/guardians of the children. All patients will receive the service-level intervention but will choose to participate in the study if they (their parents/guardians) feel they have the capacity and consent.

## Recruitment and data collection
### Inpatient mental health ward sites

Existing networks will be used to identify eligible services (wards) for the SAFER-MH/YMH studies. The study team will meet with identified services (face-to-face or remotely) to fully explain the purposes of the studies and describe their requirements. Informed consent for services to participate will be provided by the research lead/authorised person in each service, acting as 'guardian' for patients in their care. This will follow agreement with each local ward team clarifying willingness to undertake the SAFER-MH/YMH interventions. Service consent to participate in the study will be formalised through written agreements. The number of services approached, declined or considered not eligible will be recorded.

### Phase 1: SAFER-MH

We aim to recruit three pilot inpatient wards across different trusts in the northwest of England to reflect variation of NHS in England.

### Phase 2: SAFER-YMH

We aim to recruit one or two pilot inpatient wards within one trust in the northwest of England. We do not aim to recruit additional trusts because due to funding restrictions, phase 2 must be completed by March 2023.

## Patient participants

All patients on a participating ward will receive the intervention as it is a service-level change. However, we will identify patients who are willing to participate in the study assessments (questionnaires and interviews). A ward coordinator (eg, nurse or other health professionals) working within the participating services will be asked to coordinate handing out questionnaire packs to eligible patients. Potentially eligible patients will be assigned an individual study ID and given an invitation pack close to their expected discharge. This includes an invitation letter, patient information sheet (PIS) and a consent-to-contact form.

On receipt of the invitation pack, the potential participant will complete the included documents and return these to the research practitioner on-site. Consent for researchers to contact the potential participant regarding the study will be obtained, in line with the definition outlined in Article 4 (11) of the General Data Protection Regulation (GDPR) guidance: 'any freely given, specific, informed and unambiguous indication of the data subject's wishes by which he or she, by a statement or by a clear affirmative action, signifies agreement to the processing of personal data relating to him or her' (European Union, 2016). Consent to be contacted with a study recruitment pack will therefore be implied by completion and return of a completed consent-to-contact form.

### Assent for young patient participants

Assent will be sought from children/young persons' parents/guardians, or others who are not legally empowered to give consent in phase 2.

### Carer/parent/guardian participants

A study pack will be handed to each patient at discharge; this will include a carer pack, which they can choose to pass on or discard. The pack will include details of the study, paper questionnaires and a paper consent-to-contact form for interviews, as well as an online link to assess the documents remotely. For patients who agree to be contacted for an interview, we will ask again if they would like to share the information pack and contact details with their informal carer when contacted.

### Professional participants

Staff participants will be those working directly with patient participants on participating mental health wards (ie, nurses, healthcare assistants and doctors). Staff will self-identify to be involved in interviews and questionnaires through posters on the ward and emails sent by the trust research clinician/gatekeeper containing the study information pack (PIS, invitation letter, consent-to-contact form). All staff who complete the training will be given the opportunity to complete the evaluation survey.

### Procedure

#### Pre-intervention (baseline) phase

This phase will consist of the usual care that the patients in the inpatient wards receive and will last for 6 weeks (see table 1). We will assess this current practice by conducting interviews and by inspecting current documentation sheets. These will be completed by one on-site research practitioner. We will perform individual online interviews with the patients, mental health professionals and carers (relatives) to assess the degree of the satisfaction of the patients and carers as well as their views/thoughts regarding their current discharge plans. At the end of this phase, patients will be asked to complete a series of study questionnaires, including the Evaluating and Quantifying User and Carer Involvement in Mental Health Care Planning Patient-Reported Outcome Measure (EQUIP-PROM) Scale,[23] which measures the involvement of both the patients and carers in healthcare planning during/after discharge, the Care Transitions Measure (CTM)[24] and a five-item patient-centred scale focusing on mental health discharge outcomes that were identified as important to patients in previous work.[9] Patients, carers and professionals will also be invited to be interviewed about their experiences of usual care, via the study information pack.

### Training phase

The training will be a series of self-led short videos and self-help manuals to the pilot ward staff participating in the study. This phase will last for 2 weeks. We aim to use evaluation questionnaires following the training. Interviews in the training phase will explore whether the interventions have been fully comprehended by staff and ensure that we have enough time to solve any problems regarding the training as they arise.

### Intervention phase

This phase will last for 6 weeks during which the wards will receive the SAFER-MH and SAFER-YMH patient flow bundle (figure 1). Data regarding the adherence of the mental health professionals to using the SAFER-MH bundle will be collected. Individual online interviews (Zoom) will be conducted with patients and carers (within 3 months after discharge) and with mental health professionals (within 1 month after the completion of the 6-week intervention phase) to assess the feasibility and acceptability of the SAFER-MH bundle. Each patient/carer will only be interviewed once; professionals may be interviewed more than once in different stages of the project. At the end of this 6-week phase, patients will be asked to complete the same study scales that were administered during the baseline and which were completed by a different pool of patient participants discharged at

| Table 1 | Data collection procedures | |
|---|---|---|
| **Baseline phase (6 weeks)** | **Training phase (2 weeks)** | **Intervention phase (2 weeks)** |
| Patient/carer outcome measures (n=45): <br>► EQUIP-PROM Scale <br>► 5 items regarding general satisfaction about discharge planning <br>Interviews with professionals, carers and patients | Training evaluation questionnaires | Patient/carer outcome measures (n=45): <br>► EQUIP-PROM Scale <br>► 5 items regarding general satisfaction <br>► Care Transitions Measure <br>Staff outcome measures (n=45): <br>► AIM <br>► IAM <br>► FIM <br>► Adherence to the delivery of the SAFER patient flow bundle will be measured <br>Interviews with professionals, carers and patients |

AIM, Acceptability of Intervention Measure; EQUIP-PROM, Evaluating and Quantifying User and Carer Involvement in Mental Health Care Planning Patient-Reported Outcome Measure; FIM, Feasibility of Intervention Measure; IAM, Intervention Appropriateness Measure.

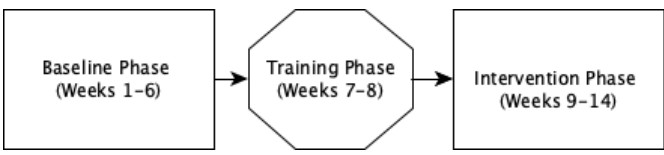

**Figure 1** Timeline of events.

the time. We will also administer the three staff outcome measures after the intervention phase is complete.

### Outcomes

The aim of the evaluation will be to determine the feasibility and acceptability of the SAFER-MH and SAFER-YMH interventions by patients, carers and professionals. Additionally, we aim to identify the completion rates and if there are any changes in the key outcomes. As this is a pre/post uncontrolled feasibility study, it will only give us a first indication about the appropriateness of the outcome measures whereas we will primarily focus on whether the SAFER-MH and SAFER-YMH is acceptable and, if so, for whom. We chose 6-week phases, as studies in UK settings indicate average adult length of stays at around 22–36 days and the NHS Mental Health Implementation Plan aims to reduce length of inpatient

psychiatric stays to a maximum of 32 days, enabling us to capture a considerable number of patients during the study.[25] [26]

### Quantitative outcomes

These will be measured in the baseline phase and intervention phase with different patients. To assess patients' and carers' perspectives, the quantitative outcomes will be measured using paper/online versions of a series of questionnaires which include: (1) the EQUIP-PROM, a 14-item co-designed measure to assess patient/carer views of care planning in mental health,[23] (2) a short four-item version of the CTM[24] and (3) the five novel items assessing the safety of mental healthcare transitions[9]; properties of these outcome measures can be found in table 2.

To assess professionals' perspectives, we will use three measures: the Acceptability of Intervention Measure, Intervention Appropriateness Measure and Feasibility of Intervention Measure. Each of the three measures contains four items to assess professionals' perspectives on a new intervention. These measures will be administered after the intervention phase only (not the baseline). Adherence to the delivery of the SAFER-MH and SAFER-YMH bundle will be measured using a purposefully

**Table 2** Properties of outcome measures used

| Validated outcome measure | Psychometric properties | Scoring system |
|---|---|---|
| EQUIP | Scale has acceptable scalability (Ho=0.69), reliability (alpha=0.92), fit to the Rasch model ($\chi2(70)=97.25$, p=0.02), and no differential item functioning or locally dependent items. | It uses a 5-point Likert scale ranging from 1='strongly disagree' to 5='strongly agree', with a middle neutral value meaning 'neither agree nor disagree'. Higher scores indicate higher-perceived quality care planning. |
| CTM-4 | The Cronbach's alpha for CTM-4 is 0.85. | The CTM items are rated on a 4-point Likert scale ranging from 1='strongly disagree' to 4='strongly agree', with a fifth response being 'don't know/ don't remember/not applicable'. The summarised score (excluding the 'not applicable' scores) is computed as the total sum divided with the number of answered items, minus 1 and multiplied by 100 to get a total score (0–100) for each respondent. Higher scores indicate better patient/carer experience of care transitions. |
| AIM, FIM, IAM | The alphas for 5-item scales were between 0.87 and 0.89. Scale refinement based on measure-specific Confirmatory Factor Analyses (CFAs) and Cronbach's alphas using vignette data produced 4-item scales (alphas from 0.85 to 0.91). A three-factor CFA exhibited acceptable fit (Comparative fit index (CFI)=0.96, Root Mean Square Error of Approximation (RMSEA)=0.08) and high factor loadings (0.75–0.89), indicating structural validity. ANOVA showed significant main effects, indicating known-groups validity. Test–retest reliability coefficients ranged from 0.73 to 0.88. | 1=completely disagree, 2=disagree, 3=neither agree nor disagree, 4=agree, 5=completely agree. Scoring instructions: scales can be created for each measure by averaging responses. Scale values range from 1 to 5. Higher scores indicate higher acceptability, appropriateness and feasibility of the intervention. |

AIM, Acceptability of Intervention Measure; ANOVA, analysis of variance; CTM-4, four-item version of the Care Transitions Measure; EQUIP, Evaluating and Quantifying User and Carer Involvement in Mental Health Care Planning; FIM, Feasibility of Intervention Measure; IAM, Intervention Appropriateness Measure.

designed self-report form, and recruitment rates will be recorded at each stage.

### In-depth qualitative outcome data

A qualitative process evaluation will be conducted using semistructured interviews to provide understanding of the study delivery and its potential scale-up. All participants who complete the questionnaires will be given the opportunity to volunteer to be interviewed. We aim to interview patients and carers at baseline and intervention, whereas professionals will be interviewed at all three stages (baseline, training and intervention). All participants (including children) will be remunerated in vouchers for participation.

### Data analysis plan

We will be following the published extension to Consolidated Standards of Reporting Trials (CONSORT)[27] when reporting the results of this feasibility study. As this is a feasibility study, no formal sample size calculation has been conducted. Our targeted sample size (n=90, phase 1; n=40, phase 2) should be large enough to consider the practicalities of recruitment and delivering the interventions.

Descriptive statistics for intervention feasibility, acceptability and appropriateness will be presented. A chart of participant flow during the whole study will be plotted (CONSORT flow chart), which will include information on the retention/withdrawal (including timing and reasons) of wards during the evaluation. We will also present the number of health practitioners who completed the questionnaires and interviews against the number of professionals working in the wards during the evaluation, and the number of patients/carers completing the questionnaires/interviews against those discharged during the duration of the study. We will also conduct basic psychometric analyses to verify the scale structure and internal consistency of the used questionnaires. For explorative purposes, we will examine differences between the three measures completed by patients recruited during the baseline phase and the measures completed by the patients during the intervention phase. Before starting with the analyses, we will examine if the data are normally distributed. If the data are normally distributed, parametric tests will be used, otherwise nonparametric tests will be used. The significance level for all analyses will be $p \leq 0.05$. The data analyst will be blinded to the phase allocation (baseline or intervention phase).

### Qualitative analysis

We will use a framework approach to qualitative analysis, to inductively code and organise the data and identify emerging themes from the interviews. The frameworks will be based on three underpinning theoretical frameworks commonly used in similar research: the Conceptual Framework for Implementation Fidelity (5 domains), Theoretical Framework of Acceptability[28] and the Theoretical Domains Framework (14 domains). The coding process will follow these steps:

1. Familiarisation (researchers immersing themselves in the data).
2. Identifying a thematic framework (to generate an index of data based on a priori questions and issues raised by participants).
3. Indexing (applying thematic framework to the data using NVivo software).
4. Charting (rearranging data in line with thematic framework).
5. Mapping and interpretation (using the charts to define concepts, map the range and nature of phenomena and find associations between themes).

### Integration of quantitative and qualitative data

The quantitative and qualitative results will be initially considered separately but we aim to examine how they converge or differ when presenting the overall conclusions.[29] Furthermore, the qualitative data collected across the two studies will be used to help explain or elaborate on the quantitative data.[29]

### Patient and public involvement

This study has a patient and public involvement service-user co-researcher who has been involved in the design and delivery of the study (AG). Patient and public involvement has been integral to this study and the development of the intervention, which has been based on a series of stakeholder consultations, workshops and consensus methods.[9 10 21 30] A core group of five patient and public involvement representatives forms a steering committee for this study.

## ETHICS AND DISSEMINATION

Ethical approval was obtained from the NHS Cornwall and Plymouth Research Ethics Committee and Surrey Research Ethics Committee (reference: 22/SW/0096 and 22/LO/0404, respectively).

The study findings will be shared with participating inpatient sites in a range of ways to engage different audiences, as well as via presentations at national and international conferences, and published in open-access peer-reviewed journals. Findings will be presented at a workshop for patients and carers, and a vlog will be created.

## DISCUSSION

This study aims to understand how to improve information sharing and capture between services during patient discharge from psychiatric hospitals to community services. Extensive stakeholder engagement with patients, carers, mental health professionals and academics has informed the intervention refinement and study co-design. We propose a pragmatic approach to test the feasibility and acceptability of a novel intervention, which

will add value to the evidence base about how information sharing/capture and patient/carer knowledge and involvement around discharge improve patient safety, healthcare utilisation and other important patient outcomes.

The findings will have to be interpreted and evaluated considering the limitations of the study design. For example, this is a pre/post uncontrolled study and the potential influence of inpatient settings cannot be controlled for through randomisation. Another limitation is that the interventions have only been developed in English and future work will need to explore how to culturally adapt it. Furthermore, some elements of the intervention may not be suitable for individuals with dementia who often occupy mental health inpatient beds[31]; however, funding has been obtained to develop this in future work. Finally, the intervention sites will be identified through collaborators, so selection bias must be considered.

The feasibility study will include paper and pdf editable versions of the intervention documents (eg, social information capture and patient written discharge plan), which have not been integrated into the trusts' electronic records yet. Understanding how the intervention documents will fit within the trusts' electronic records is a key learning task of this feasibility study, which will potentially inform a larger randomised controlled trial (RCT). Similarly, as this is a feasibility study, we are mostly interested in the completion rate of the scales, and the views by patients/carers and staff members about the relevance of these measures to their experiences in relation to hospital discharge. We would not have enough power to generate any meaningful quantitative analysis using our outcome measures and we will not collect quantitative data around readmission rates or adverse events at this stage. However, if this study produces convincing evidence that the intervention is feasible/acceptable, we will request further funding to conduct an RCT, which will involve more extensive data collection (including routinely collected data on readmissions and adverse events) and a comprehensive quantitative analysis plan for a definitive RCT. There is considerable evidence about the importance of community follow-up of patients at post-discharge. Although this is a pre-discharge intervention, there is growing evidence that continuity of care and post-discharge follow-up of patients could enhance safety. Depending on the feedback that we will obtain from the intervention recipients/deliverers during this feasibility study, we will explore whether further adaptations are needed to include post-discharge elements in the intervention protocol.

Improving safety at mental health discharge is a key concern for patients, professionals and policymakers. Therefore, interventions aiming to improve knowledge sharing, shared decision-making and involvement of patients/carers are considered preferable.[21 23 32] This study will determine the feasibility and acceptability of a multicomponent intervention to improve involvement of patients and/or carers in discharge planning. This will

provide evidence for the acceptability of the SAFER-MH and SAFER-YMH to inform future improvements to scale to an RCT, with concurrent process evaluation and economic evaluation.

**Author affiliations**
[1]NIHR Greater Manchester Patient Safety Translational Research Centre, University of Manchester, Manchester, UK
[2]NIHR School for Primary Care Research, University of Manchester, Manchester, UK
[3]Department of Primary Care and Mental Health, University of Liverpool, Liverpool, UK
[4]Centre for Pharmacoepidemiology and Drug Safety Research, University of Manchester, Manchester, UK
[5]Centre for Primary Care, University of Manchester, Manchester, UK
[6]Division of Nursing, Midwifery & Social Work, University of Manchester, Manchester, UK
[7]Centre for Suicide Prevention, University of Manchester, Manchester, UK
[8]Manchester Centre for Health Psychology, University of Manchester, Manchester, UK

**Acknowledgements** We would like to thank all of the patient, public, carer and professional stakeholders who have been involved in the co-design and development of the SAFER-MH intervention to date.

**Contributors** NT and MP had the initial research idea and obtained funding for this study. NT, MP and IA formulated the research questions and designed the study. AH, NT, IA and MP devised the qualitative and statistical analysis plan. NT, MP and IA drafted the protocol. AG led the patient and public involvement components of the study design. JL drafted the protocol and completed ethics documentation/application. RNK, CP, SJG, NK, CA, TB, SMC and CR substantially contributed to the manuscript by providing review comments and edits and approved the final manuscript.

**Funding** This work is supported by the NIHR Greater Manchester Patient Safety Translational Research Centre (grant number: PSTRC-2016-003) and NIHR School for Primary Care Research Capacity award 21/22 (grant number C015; postdoctoral fellowship award).

**Disclaimer** The views expressed are those of the author(s) and not necessarily those of the NIHR or the Department of Health and Social Care.

**Competing interests** None declared.

**Patient and public involvement** Patients and/or the public were involved in the design, or conduct, or reporting, or dissemination plans of this research. Refer to the Methods section for further details.

**Patient consent for publication** Not required.

**Provenance and peer review** Not commissioned; externally peer reviewed.

**ORCID iD**
Natasha Tyler http://orcid.org/0000-0001-8257-1090

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
