## [Reviewer comments · BMJ Open]

ARTICLE DETAILS

TITLE (PROVISIONAL)	Protocol for a non-randomised feasibility study evaluating a co-designed care bundle to improve patient safety at discharge from adult and adolescent mental health services (SAFER-MH and SAFER-YMH)
AUTHORS	Tyler, Natasha; Angelakis, Ioannis; Keers, Richard; Planner, Claire; hodkinson, alexander; Giles, Sally; Grundy, Andrew; Kapur, Navneet; Armitage, Chris; Blakeman, Tom; Campbell, Stephen; Robinson, Catherine; Leather, Jessica; Panagioti, Maria

VERSION 1 – REVIEW

REVIEWER	Chakraborty, Nandini Leicestershire Partnership NHS Trust
REVIEW RETURNED	05-Dec-2022

GENERAL COMMENTS	While I think that this is a very clearly written protocol with a defined question and well-meant outcomes, on a practical level as a clinician, there are numerous gaps which need addressing for this study to have an impact on the safety of discharges. The suggested interventions at admission, weekly tasks and discharge are clearly written and if implemented would lead to good practice. In a well performing inpatient unit, several if not all of these possibly take place anyways. The main gaps in communication between inpatient units and community follow up which make a discharge unsafe, happen for several reasons which have not been addressed in this protocol. As a full-time clinician, these are the areas I would highlight: 1. The authors have mentioned that 3 pilot wards will be included in the study and all eligible, capacious, consenting patients will be included. What happens in a situation where patients change wards for bed management reasons rather than clinical reasons? That is out of the control of this study. Due to acute bed shortages, patients are often, less than ideally, moved around to accommodate more acute admissions. They are sent on leave, sent out of area, or have unplanned discharges. There is no mention or consideration of this in the study protocol.2. The use of electronic records can be a useful method in making sure staff in different teams of an NHS Trust have access to the same information. There is no mention of electronic systems, how these can be best used or changed in order to make communication more robust and discharge safer.3. There is wide use of locum doctors and bank staff in a time when we are facing an acute crisis of stable staffing in the NHS. The safety which comes with a patient having the same consultant psychiatrist and the same named nurse in hospital, following on to a consistent CPN and a consistent community psychiatrist, is lost within numerous staff changes which no amount of paperwork can
--

	replace. 4. There needs to be a mention about involving community staff while patients are admitted. There is a mention of multiagency and MDT meetings. How these can look when a patient is completely new to services (needing referrals) or when they already have a CMHT involved, need to be clarified. 5. The outcomes both qualitative and quantitative are based on perspectives of patients, professionals and carers. Are there any plans for objective quantitative data for example readmission rates and adverse events? 6. A patient needs to be reviewed in the community in a safe period of time after discharge. The protocol does not include a maximum timeframe for a first contact with clinician in the community which is part of safe discharge in many Trusts (7-day follow up). I see this study protocol as suggesting more paperwork for staff (to be evidenced), senior reviews, MDT meetings, discharge planning etc. But without consistent staffing, it becomes meaningless paperwork. I acknowledge the hard work and good intent which has gone into this protocol, and the suggested interventions are around good practice. However, what has been making discharges unsafe, what is leading to the missed communication, has not been dealt with at all in the protocol, in my opinion. Without consistency in named clinicians, without a protocol of how to deal with a bed crisis without moving patients around and admitting them out of area, any protocol is bound to fail.
--	---

REVIEWER	Barbato, Angelo
	IRCCS-Istituto di Ricerche Farmacologiche Mario Negri
REVIEW RETURNED	23-Dec-2022

GENERAL COMMENTS	I have two important remarks to your protocol: 1. The timeline of the study is unclear. First, you have to clarify whether the study will be entirely conducted during the inpatient stay of the recruited sample. The intervention phase is supposed last 6 weeks. To this regard, a number of questions arise: what will you do in case of early or unplanned discharge ? Do you have data about the average length of stay in the wards of the area selected for the sample recruitment ? 2. For the patient outcome assessment, you will use tools such as the EQUIP-PROM and the CTM-4. However, these are not widely known tools, therefore you should provide information on their psychometric properties and the scoring system. Do will use them to produce categorical data (i.e. satisfied vs unsatisfied) or continuous variables (i.e. mean satisfaction scores) ? How will the define the positive and negative outcomes ? 3. For the professional outcome assessment you will use three mesures of acceptability, appropriateness and feasibility of Intervention. However you have to provide information on the tools used for the measurement and the definition of outcomes in all three dimensions. The above remarks have to be addressed in a revised version of your paper, to consider it for publication.
--

VERSION 1 – AUTHOR RESPONSE

Reviewer: 1 Dr. Nandini Chakraborty, Leicestershire Partnership NHS Trust Comments to the Author:	
While I think that this is a very clearly written protocol with a defined question and well-meant outcomes, on a practical level as a clinician, there are numerous gaps which need addressing for this study to have an impact on the safety of discharges. The suggested interventions at admission, weekly tasks and discharge are clearly written and if implemented would lead to good practice. In a well performing inpatient unit, several if not all of these possibly take place anyways. The main gaps in communication between inpatient units and community follow up which make a discharge unsafe, happen for several reasons which have not been addressed in this protocol. As a full-time clinician, these are the areas I would highlight:	Thank you for your comments
1. The authors have mentioned that 3 pilot wards will be included in the study and all eligible, capacitous, consenting patients will be included. What happens in a situation where patients change wards for bed management reasons rather than clinical reasons? That is out of the control of this study. Due to acute bed shortages, patients are often, less than ideally, moved around to accommodate more acute admissions. They are sent on leave, sent out of area, or have unplanned discharges. There is no mention or consideration of this in the study protocol.	Only participants who are discharged from the ward in question will be given the option to be involved in data collection, as a key component is the patient written discharge plan. This has been described in the participants section on page 5.
2. The use of electronic records can be a useful method in making sure staff in different teams of an NHS Trust have access to the same information. There is no mention of electronic systems, how these can be best used or changed in order to make communication more robust and discharge safer.	We have added this into the limitations section (page 12), as a small-scale feasibility study the documents will not be integrated into electronic systems, but we hope that learning from this study will enable this to happen or be considered for a larger RCT in the future.
3. There is wide use of locum doctors and bank staff in a time	Lack of continuity of care is a key safety concern. Continuity of

when we are facing an acute crisis of stable staffing in the NHS. The safety which comes with a patient having the same consultant psychiatrist and the same named nurse in hospital, following on to a consistent CPN and a consistent community psychiatrist, is lost within numerous staff changes which no amount of paperwork can replace.	care has been raised while we were co-developing this intervention with mental health professionals using RAND/UCLA methods . Although professionals agreed about the importance of continuity of care, there were concerns about its feasibility in the current state of the NHS. We have added a limitation in the discussion where we suggest that continuity of care is an important contributor to patient safety but it is hard to implement and sustain given the staffing crisis in NHS. Despite this, depending on the feedback that we will obtain by the intervention recipients/deliverers during this feasibility study, we will explore whether further adaptations are needed to include post-discharge elements in the intervention protocol (see page 11, last paragraph).
4. There needs to be a mention about involving community staff while patients are admitted. There is a mention of multiagency and MDT meetings. How these can look when a patient is completely new to services (needing referrals) or when they already have a CMHT involved, need to be clarified.	The social information capture form prompts ward staff to contact any relevant agency (i.e. community staff/social services) when important social information is missing. We hope the intervention will encourage greater communication at admission and throughout stay, but this feasibility study will enable us to see if this has worked in practice.
5. The outcomes both qualitative and quantitative are based on perspectives of patients, professionals and carers. Are there any plans for objective quantitative data for example readmission rates and adverse events?	As this is a feasibility study, we could not conduct quantitative analysis with enough power to generate any meaningful results. If this intervention is proved to be feasible and acceptable and provided that we received further funding, we will undertake a a larger RCT which will collect data on adverse events and readmissions . We have added this into the limitations section on page 12.
6. A patient needs to be reviewed in the community in a safe period of time after discharge. The protocol does not include a maximum timeframe for a first contact with clinician in the community which is part of safe discharge in many Trusts (7-day follow up). I see this study protocol as suggesting more paperwork for staff (to be evidenced), senior reviews, MDT meetings, discharge planning etc. But without consistent staffing, it becomes meaningless paperwork.	Thank you, this is a very helpful point. We will be doing interviews with staff members, patients and carers during this feasibility study. If staff members, patients/carers agree that adding a 7-day maximum timeframe for a first contact with a clinician in the community, we will make this addition in the forms. We partly agree with the reviewer that consistent staffing is important but each ward works differently. During our RAND/UCLA study, mental health professionals who were working in wards made it clear that the responsible staff need to be named/agreed by each ward rather than being pre-prescribed as part of the intervention protocol.
I acknowledge the hard work and good intent which has gone into this protocol, and the suggested interventions are around good practice. However, what has been making discharges unsafe, what is leading to the missed communication, has not been dealt with at all in the protocol, in my opinion. Without consistency in named clinicians, without a protocol	Although we accept these criticisms, the suggestions of the reviewer would require in depth system changes and considerable financial investments. We wanted to test the feasibility/acceptability a low cost best practice intervention which does not require a huge time/resource investment. If this intervention appears to be feasible/acceptable, we will consider adding further elements to it such as continuity of care elements or stricter guidance about named clinicians.

of how to deal with a bed crisis without moving patients around and admitting them out of area, any protocol is bound to fail.	
Reviewer: 2 Dr. Angelo Barbato, IRCCS- Istituto di Ricerche Farmacologiche Mario Negri Comments to the Author: I have two important remarks to your protocol:	Thank you for your comments
1. The timeline of the study is unclear. First, you have to clarify whether the study will be entirely conducted during the inpatient stay of the recruited sample. The intervention phase is supposed last 6 weeks. To this regard, a number of questions arise: what will you do in case of early or unplanned discharge ? Do you have data about the average length of stay in the wards of the area selected for the sample recruitment ?	We chose a 6 week phase, as studies in the UK settings indicate that the average adult length of stays are around 22-36 days (young people's length of stay is around 6 weeks based on our discussion with wards) and The NHS Mental Health Implementation Plan aims to reduce length of inpatient psychiatric stays to a maximum of 32 days, enabling us to capture a considerable number of patients during the study. Our intervention is applicable even if patients are discharged before the 6 week-period. All intervention components can be implemented within 2 weeks, which is the suggested minimum length of stay in mental health hospitals that most of our stakeholders reported. The opposite might be a problem e.g. patients staying in hospital longer than our proposed 6 week phase. However, the purpose of this feasibility study is not to evaluate the intervention but to answer these practical questions on how to evaluate this intervention in the future, e.g. is our study duration or measures optimal for a larger evaluation study. We have added more description around this to the outcomes section of the methods see page 8 and participants section on page 5.
2. For the patient outcome assessment, you will use tools such as the EQUIP-PROM and the CTM-4. However, these are not widely known tools, therefore you should provide information on their psychometric properties and the scoring system. Do will use them to produce categorical data (i.e. satisfied vs unsatisfied) or continuous variables (i.e. mean satisfaction scores) ? How will the define the positive and negative outcomes ?	We have created a new table to highlight the psychometric properties and the scoring system of these scales. We will mostly produce continuous variables and we will follow the scoring guidelines for each scale. However, as this is a feasibility study we are mostly interested in the completion rate of the scales, and views by patients/carers and staff members about the relevance of these measures to their discharge experience. We have recently completed a number of systematic reviews of discharge interventions, which showed that most interventions to date only capture hard outcomes such as readmissions but fail to capture the perspectives of patients, carers and professionals. Although these measures have not been used widely, we wanted to see if it is feasible to use them in future evaluations See table 2. https://pubmed.ncbi.nlm.nih.gov/31760955/ https://jamanetwork.com/journals/jamanetworkopen/fullarticle/2791849
3. For the professional outcome assessment you will use	We have described the psychometric properties and the scoring system of the measures of acceptability,

three measures of acceptability, appropriateness and feasibility of Intervention. However you have to provide information on the tools used for the measurement and the definition of outcomes in all three dimensions. The above remarks have to be addressed in a revised version of your paper, to consider it for publication.	appropriateness and feasibility of Intervention in the table above
--	---

VERSION 2 – REVIEW

REVIEWER	Barbato, Angelo IRCCS-Istituto di Ricerche Farmacologiche Mario Negri
REVIEW RETURNED	03-Feb-2023
GENERAL COMMENTS	I am satisfied by the changes introduced in the paper